# Human-specific evolutionary markers linked to foetal neurodevelopment modulate brain surface area in schizophrenia

Maria Guardiola-Ripoll [1,2✉], Carmen Almodóvar-Payá [1,2], Angelo Arias-Magnasco[1], Mariona Latorre-Guardia[1], Sergi Papiol[2,3,4], Erick J. Canales-Rodríguez [2,5], María Ángeles García-León[1,2], Paola Fuentes-Claramonte[1,2], Josep Salavert [6], Josep Tristany[7], Llanos Torres[8], Elena Rodríguez-Cano [1,2,9], Raymond Salvador[1,2], Edith Pomarol-Clotet[1,2,11] & Mar Fatjó-Vilas [1,2,10,11✉]

Schizophrenia may represent a trade-off in the evolution of human-specific ontogenetic mechanisms that guide neurodevelopment. Human Accelerated Regions (HARs) are evolutionary markers functioning as neurodevelopmental transcription enhancers that have been associated with brain configuration, neural information processing, and schizophrenia risk. Here, we have investigated the influence of HARs' polygenic load on neuroanatomical measures through a case-control approach (128 patients with schizophrenia and 115 controls). To this end, we have calculated the global schizophrenia Polygenic Risk Score (Global PRS$_{SZ}$) and that specific to HARs (HARs PRS$_{SZ}$). We have also estimated the polygenic burden restricted to the HARs linked to transcriptional regulatory elements active in the foetal brain (FB-HARs PRS$_{SZ}$) and the adult brain (AB-HARs PRS$_{SZ}$). We have explored the main effects of the PRSs and the PRSs x diagnosis interactions on brain regional cortical thickness (CT) and surface area (SA). The results indicate that a higher FB-HARs PRS$_{SZ}$ is associated with patients' lower SA in the lateral orbitofrontal cortex, the superior temporal cortex, the pars triangularis and the paracentral lobule. While noHARs-derived PRSs show an effect on the risk, our neuroanatomical findings suggest that the human-specific transcriptional regulation during the prenatal period underlies SA variability, highlighting the role of these evolutionary markers in the schizophrenia genomic architecture.

[1] FIDMAG Germanes Hospitalàries Research Foundation, Barcelona, Spain. [2] CIBERSAM (Biomedical Research Network in Mental Health; Instituto de Salud Carlos III), Madrid, Spain. [3] Institute of Psychiatric Phenomics and Genomics (IPPG), University Hospital, LMU Munich, Munich, Germany. [4] Max Planck Institute of Psychiatry, Munich, Germany. [5] Signal Processing Laboratory 5 (LTS5), École Polytechnique Fédérale de Lausanne (EPFL), Lausanne, Switzerland. [6] Hospital Sant Rafael, Germanes Hospitalàries, Barcelona, Spain. [7] Hospital Sagrat Cor, Germanes Hospitalàries, Martorell, Spain. [8] Hospital Mare de Déu de la Mercè, Germanes Hospitalàries, Barcelona, Spain. [9] Hospital Benito Menni, Germanes Hospitalàries, Sant Boi de Llobregat, Spain. [10] Departament de Biologia Evolutiva, Ecologia i Ciències Ambientals, Universitat de Barcelona, Barcelona, Spain. [11]These authors jointly supervised this work: Edith Pomarol-Clotet, Mar Fatjó-Vilas. ✉email: mguardiola@fidmag.org; mfatjo-vilas@fidmag.org

Schizophrenia (SZ) is a complex neuropsychiatric disorder characterised by symptoms that alter perception and behaviour, such as hallucinations, delusions, and affectations of higher-order cognitive functions. These symptoms intimately relate the disorder with traits distinguishing humans as a species, such as abstraction, language, thinking, and some aspects of social cognition, such as Theory of Mind[1–3]. Accordingly, while the foundations of this multifactorial and complex disorder are not entirely understood, multiple pieces of evidence straightforwardly point towards a neurodevelopmental and evolutionary origin.

On the one hand, the polygenic architecture of SZ, with thousands of genetic variants with additive effects accounting up to 80% of its heritability[4–7], converges with pathways related to developmental, neuronal, and synaptic differentiation mechanisms[8–10]. Such pathways are tightly orchestrated and involve gene expression regulatory mechanisms of paramount importance for brain ontogeny[11], which, at the same time, are highly influenced by the environmental events occurring during prenatal and early periods. Remarkably, prenatal and obstetric complications are associated with an increased risk for SZ[12], and their presence correlates with a higher genomic risk for the disorder[13]. In parallel, the placenta-associated genomic burden for schizophrenia is linked to reduced brain volumes in neonates and poorer cognitive development during the first two years of life[14], while delayed developmental milestones are associated with the disorder and predict psychotic symptoms in childhood and adulthood[15–17]. Also, highly heritable neuroanatomical measures, such as cortical thickness (CT) and cortical surface area (SA), show differences between patients with SZ and healthy controls, even prior to the onset of the psychotic symptoms[18,19]. All this evidence sustains the prevailing aetiological hypothesis that SZ results from environmental and genetic interactions modulating and deviating neurodevelopmental trajectories during the intrauterine and perinatal periods as well as during childhood and early adolescence[20,21] that disrupt the ontogenetic plan guiding brain architecture, brain configuration, and brain functioning.

On the other hand, the common prevalence of the disorder (nearly 1%)[22–24], and the fact that people affected, particularly males, have a reduced rate of reproduction (fitness) compared with non-affected individuals[25,26], raise the question of why the genetic variants that increase the likelihood of suffering from SZ have persisted in the genetic pool. This, together with the close relationship between several clinical aspects of the disorder and human-specific cognitive traits[1], has boosted the evolutionary view of the disorder. Accordingly, the evolutionary hypothesis of SZ suggests that the disorder emerged as a costly trade-off in the evolution of the ontogenetic mechanisms guiding human-specific neurodevelopment and sustaining complex cognitive abilities[27–30].

While the evolutionary traces of SZ are challenging to follow, studying human-specific genomic changes through comparative genomics may lead to a better comprehension of human-specific phenotypic traits and increased knowledge of what genetic changes contributed to making us human[31–33]. In this sense, Human Accelerated Regions (HARs) might be helpful. HARs are evolutionary conserved genomic regions that have experienced significant changes after human and chimpanzee divergence[34–39]. This accelerated evolution is suggested to reflect HARs' role in some human-specific characteristics. Most HARs are intergenic, within introns near protein-coding genes, transcription factors and DNA-binding proteins[40–44]. All the studies that intended to characterise HARs' functional role converge in highlighting them as transcription factors binding sites, transcription factors on their own and participants in the neurodevelopmental gene expression machinery[43,45–47].

Recently, studies inspecting the expression patterns of HARs-associated genes (HARs-genes) show their implication in human-specific cortical expansion, brain functional connectivity and brain's neural information processing. First, a comparative study exploring the cortical expansion in humans and chimpanzees described that the expression profiles of HARs-genes correlate with the expansion of higher-order cognitive networks, such as the frontoparietal and the default mode networks[48]. The same study reveals that the genetic variability in HARs-genes expressed in the brain (HARs-brain genes) is associated with the default mode network functional variation in healthy subjects[48]. Second, the expression patterns of HARs-brain genes have been related to individual variability in functional connectivity and the brain's information processing[49,50]. Notably, these studies report that HARs-brain genes show the highest expression in higher-order cognitive networks, such as the frontoparietal and the default mode networks, the ones with the greatest functional heterogeneity across individuals and with predominant synergistic interactions[49,50].

The evidence on HARs' contribution to human-specific brain architectural configuration, functioning and information processing is also accompanied by studies that describe that HARs' genetic variability influences the risk for SZ. For example, the investigation into the overlap between HARs and whole-genome common genetic variants shows that the SZ polygenic background is enriched in genes associated with these evolutionary regions[51]. In line, subsequent findings also described that Single Nucleotide Polymorphisms (SNPs) in HARs or in linkage disequilibrium with them are more likely associated with the disorder[52]. Notwithstanding, to our knowledge, HARs modulation of brain measures in SZ has been scarcely explored, and further studies using brain-based phenotypes to assess their role in the disorder are necessary.

Considering the polygenic nature of both cortical structural configuration and SZ's susceptibility[10,53], the use of measures summarising this complex genetic architecture, such as Polygenic Risk Scores (PRSs), would be helpful to disentangle the genetic roots not only of the disorder but of complex brain traits. The PRS is a quantitative measure of the individual genetic burden for a trait based on data from Genome-Wide Association Studies (GWAS). Although PRS methods do not yet provide clinically feasible information in psychiatric disorders[54], schizophrenia PRSs are highly informative for assessing the individual risk for the disorder at the research level and are highly consistent across studies and samples[55]. PRSs can be calculated on a genome-wide basis, but also within subsets of SNPs defined according to their affiliation to specific biological pathways of interest[56]. Therefore, based on the evidence of HARs' role in neurodevelopment, brain configuration and susceptibility for SZ, we aimed to investigate the modulatory effect of HARs' polygenic load on neuroanatomical measures through a neuroimaging genetics approach in healthy controls and patients with SZ.

We generated different PRSs summarising HARs' genetic variability, specifically including HARs SNPs related to active regulatory elements in the foetal and adult brain. We explored whether the PRSs modulated CT and SA differently depending on the health/disease condition. Our findings point to a specific effect of HARs linked to foetal brain regulatory elements on patients' cortical SA, emphasising the importance of human-specific changes in early neurodevelopment and in the structural changes associated with the disorder.

## Results

**Case-control PRS comparison.** Four SZ-based PRSs were estimated: one summarizing the global schizophrenia polygenic burden (Global PRS$_{SZ}$), another specific to HARs (HARs PRS$_{SZ}$), as well as two PRSs specific to HARs linked to transcriptional regulatory elements active in the foetal (FB-HARs PRS$_{SZ}$) and adult

**Table 1 Polygenic risk score (PRS) comparisons between healthy controls (HC) and patients with schizophrenia (SZ).**

|  | HC mean (sd) | SZ mean (sd) | β (se) | W | $R^2$ | FDR-pval |
|---|---|---|---|---|---|---|
| Global $PRS_{SZ}$ | −201.14 (1.23) | −200.10 (1.02) | 0.79 (0.14) | 32.97 | 0.20 | $3.75 \times 10^{-8*}$ |
| HARs $PRS_{SZ}$ | −5.45 (0.17) | −5.41 (0.19) | 1.32 (0.73) | 3.28 | 0.02 | $9.33 \times 10^{-2}$ |
| FB-HARs $PRS_{SZ}$ | −0.53 (0.05) | −0.52 (0.04) | 2.07 (2.96) | 0.49 | 0.003 | $4.85 \times 10^{-1}$ |
| AB-HARs $PRS_{SZ}$ | −0.43 (0.04) | −0.42 (0.04) | 7.69 (3.60) | 4.55 | 0.02 | $6.57 \times 10^{-2}$ |

Means and standard deviations (sd) are given for the four estimated PRSs (Global $PRS_{SZ}$, HARs $PRS_{SZ}$, FB-HARs $PRS_{SZ}$, AB-HARs $PRS_{SZ}$), separately for healthy controls (HC) and patients with schizophrenia (SZ). The logistic regression statistics include the β and standard error (se), the Wald (W), the Nagelkerke's pseudo-$R^2$ ($R^2$), and the adjusted p-values after FDR correction (FDR-pval).
*Significant findings at FDR-pval < 0.001.

brain (AB-HARs $PRS_{SZ}$). Case-control comparisons revealed differences in the Global $PRS_{SZ}$: patients presented higher SZ genetic load as compared to HC. The HARs-derived PRSs did not show between-group differences (Table 1). No significant PRS effect was detected on patients' clinical profiles (age at onset, illness duration, Positive and Negative Syndrome Scale (PANSS) scores, and antipsychotic medication, given as Chlorpromazine (CPZ) equivalents in mg/day) (Supplementary Data 1).

**PRS associations with cortical thickness and surface area measures**. When examining the extent to which the four PRSs modulated CT and SA measures of each cortical region, we detected that no PRS estimate influenced the CT in HC or patients with SZ. Along the same line, the PRSs x diagnosis interactions on CT did not evidence significant effects (Supplementary Data 2).

In contrast, the linear regression analyses revealed that among patients, the FB-HARs $PRS_{SZ}$ significantly affected the cortical SA in four regions of the right hemisphere: the lateral orbitofrontal cortex (β = −1581.796, Standardised β = −0.234, SE = 440.443, adjusted $R^2$ = 0.491, FDR-pval = 0.008), the superior temporal cortex (β = −1960.151, Standardised β = −0.235, SE = 545.898, adjusted $R^2$ = 0.488, FDR-pval = 0.008), the pars triangularis (β = −1409.886, Standardised β = −0.242, SE = 438.928, adjusted $R^2$ = 0.322, FDR-pval = 0.020) and the paracentral lobule (β = −842.876, Standardised β = −0.233, SE = 282.910, adjusted $R^2$ = 0.264, FDR-pval = 0.031) (Fig. 1). A higher SZ risk load in FB-HARs was associated with lower SA values in these regions. Conversely, no other associations were observed between the other estimated PRSs and SA, neither in HC nor in interaction with diagnosis (Supplementary Data 2). The SA variability in these four regions of interest was not related to the clinical profiles of patients (Supplementary Data 3).

Inspecting the genomic context of the SNPs in FB-HARs $PRS_{SZ}$ showed that 54.1% were in intergenic regions, 26.5% in introns and 16.6% in intronic non-coding RNA. The SNPs were mapped into 223 genes (Supplementary Data 4). The subsequent functional annotation results highlighted that these genes were enriched in several Gene Ontology (GO) categories (Fig. 2, Supplementary Data 5). The biological processes more significantly enriched were neuron differentiation, neurogenesis, neuron development, and head development. Also, the only cellular component category enriched was cell junction.

## Discussion

Through a neuroimaging genetic approach, we have evaluated the SZ's polygenic burden of human-specific evolutionary markers such as HARs on brain-based phenotypes closely related to the pathophysiology of the disorder. Although studies on HARs are an emerging field, as a recent review by our group highlights[57], and the expression profiles of HARs-brain genes have been studied in relation to brain structural changes across different psychiatric disorders[48], our study is the first to assess the effect of HARs genetic

variability on brain cortical measures in patients with SZ and healthy controls. These analyses provide evidence of the impact of foetal active regulatory HARs on the cortical surface area of different brain regions in patients with SZ. These findings highlight the importance of human-specific genetic changes guiding human brain cortical architecture, particularly those affecting the gene-expression machinery active during prenatal stages.

The comparisons between the different PRSs estimates across diagnostic groups show that individuals with a diagnosis of SZ present higher Global $PRS_{SZ}$ than healthy individuals, and thus, a higher polygenic load for the disorder. This result aligns with the current view on the value of the $PRS_{SZ}$ as a highly informative genetic vulnerability marker for its consistency across numerous studies, not only limited to comparisons between patients and controls[58,59], but also through family approaches, which show the intermediate genetic load that healthy relatives of affected patients have[60,61].

However, in our sample, the analyses indicate no significant differences across diagnostic groups when using the HARs-related $PRS_{SZ}$. The lack of association between HARs-derived PRSs and SZ risk in our sample may reflect the broader genotypic background underlying the diagnosis, but it could also result from insufficient power. Therefore, future association studies with larger case-control samples are needed to deepen the direct link between HARs' polygenic load and the risk of developing SZ. It is challenging to directly compare our findings with previous research because of the absence of HARs-based PRS studies. Some previous SZ association research exploring the variability in candidate HARs points towards a link between these evolutionary regions and the risk for the disorder. For instance, the haplotypic variability at the *HAR1A* gene, a novel RNA gene with a presumable neurodevelopmental role that harbours the HAR with the highest substitution rate in humans as compared to chimpanzees[44], was associated with auditory hallucinations in patients with a schizophrenia-spectrum disorder in a European sample[62]. Also, several candidate HARs-SNPs, altering transcription factor binding sites and presenting methylation marks of active promoters, repressors, or enhancers in the brain, were associated with the risk for SZ and modulated cognitive performance in a north-Indian population[63,64]. Convergently, genome-wide-based studies also describe that HARs-genes and HARs-brain genes are associated with SZ at the GWAS level[48,51,52,65]. Some of these studies, indeed, went beyond common variability and showed that rare variants in HARs-genes were also enriched with the disorder[48]; suggesting, therefore, that joint analysis of common and rare variants can help disentangling the role of HARs variability on the susceptibility for the disorder and its specific phenotypes.

In our investigation on the contribution of HARs polygenic background on cortical neuroanatomical measures, we report a modulatory effect of FB-HARs $PRS_{SZ}$ on the SA within patients. These findings suggest that the genetic variability in HARs associated with regulatory elements uniquely active in the foetal brain would specifically influence brain phenotypes in SZ. Results

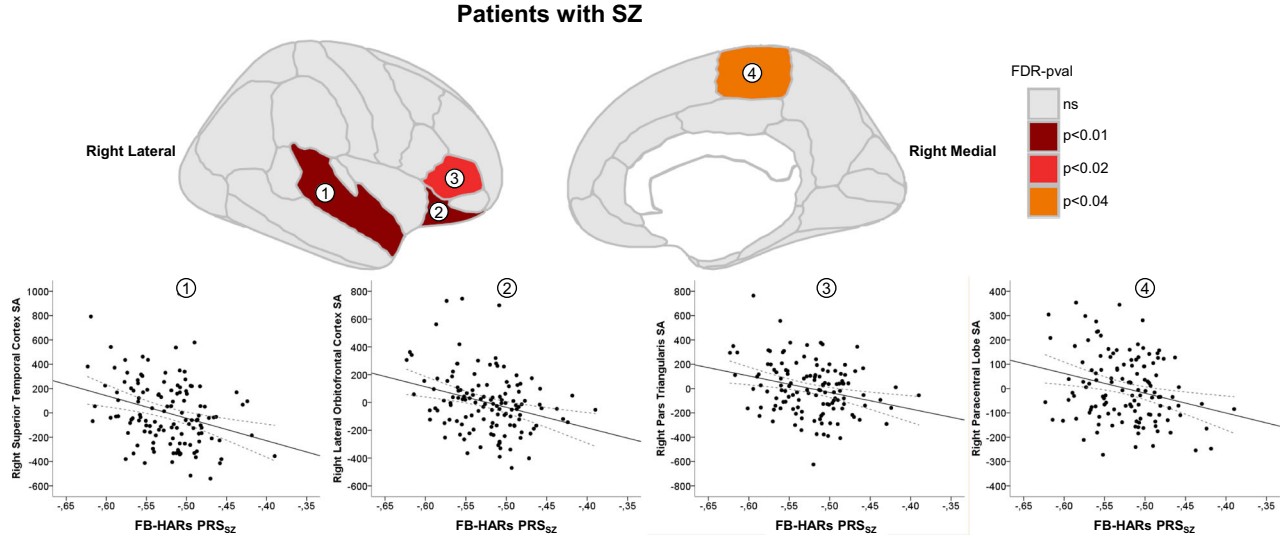

**Fig. 1 Brain regions and scatter plots with significant FB-HARs PRS$_{SZ}$ effect on surface area in patients with schizophrenia (SZ).** Brain plots include the lateral and medial sagittal views for the right hemisphere. The coloured regions are the ones with significant FB-HARs PRS$_{SZ}$ effect on surface area (SA). Within patients ($n = 128$), the scatter plots show the FB-HARs PRS$_{SZ}$, on the $X$-axis, and the unstandardised SA residuals (estimated regressing out the covariates), on the $Y$-axis. These plots evidence the negative correlation between the two measures (black solid line and black dashed lines representing the regression line and the 95% confidence intervals, respectively). Each region is numerically labelled as follows: (1) superior temporal cortex; (2) lateral orbitofrontal cortex; (3) pars triangularis; and (4) paracentral lobule.

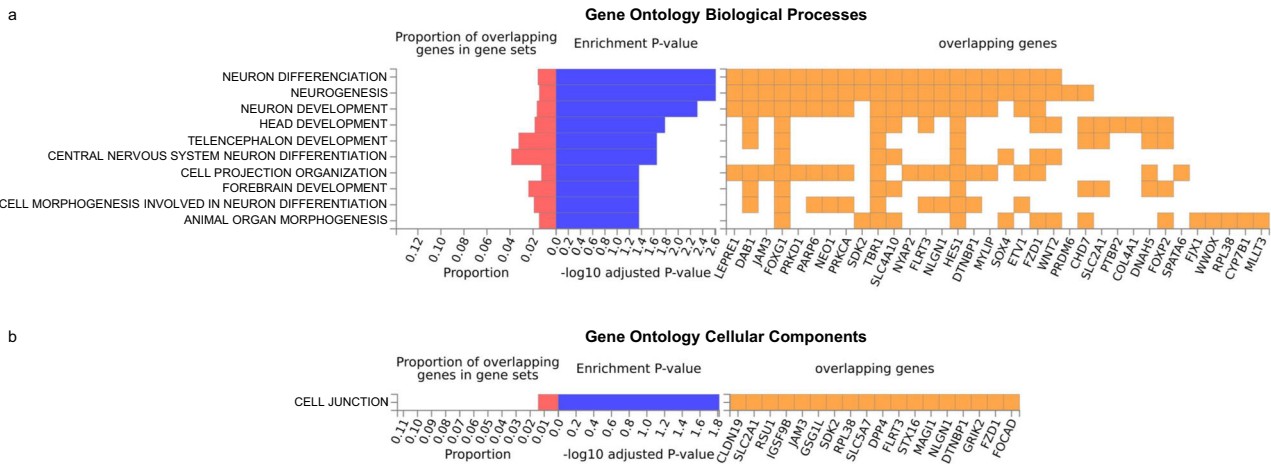

**Fig. 2 Gene set enrichment results.** Functional annotation results derived from FUMA (https://fuma.ctglab.nl/) in Gene Ontology (GO) categories for biological processes (**a**), and cellular components (**b**). In each panel, the GO categories can be seen on the left, followed by the proportion of the overlapping genes, the associated *p*-value after FDR correction (Enrichment *P*-value), and the overlapping genes on the right.

show that as the FB-HARs PRS$_{SZ}$ increases (i.e. as more SZ risk load accumulates in these HARs associated with foetal active regulatory elements), patients present lower cortical SA in the lateral orbitofrontal cortex, the superior temporal cortex, the pars triangularis and the paracentral lobe. First, these findings converge with data highlighting the importance of the foetal period (and the gene expression circumscribed to it) to understand the neurodevelopmental trajectories linked to SZ[14]. Also, our findings align with the ENIGMA Consortium study reporting widespread smaller surface area in SZ, with the largest effect sizes in the frontal and temporal lobe regions[19]. In spite of the detected differences in SZ, the reductions in SA were not associated with the patients' symptoms, the onset of the disorder or its duration. These outcomes are aligned with the findings from van Erp et al., that reported that the regional SA changes in patients were not correlated with PANSS total, positive, and negative symptom scores[19]. Even so, in previous studies, SA reductions in the right

frontal cortex have been correlated with more severe negative symptoms in patients with psychosis, including SZ, schizoaffective disorder, and bipolar disorder type I with psychotic symptoms[66]. In the same line, other structural and functional alterations in the superior temporal region have been reported in relation to formal thought disorder in patients[67,68].

From an evolutionary perspective, if we focus on the regions significantly modulated by the SZ genetic burden in active foetal brain HARs, we can draw attention to the orbitofrontal region and the temporal cortex. These are among the regions suffering the largest expansion in the human cortex in comparison with chimpanzees[48]. Wei et al.[48] described that those areas of the orbital frontal gyrus and the temporal lobe experienced an x4 and x3 expansion, respectively, and evidenced that the transcription profile of 1,711 HARs-genes positively correlated to the pattern of human cortical expansion, meaning that the highest HARs-gene expression occurs in highly expanded areas of the human cortex.

Moreover, it has been proposed that the anatomical changes in the human temporal cortex compared to other primates may be involved in higher-level cognitive functions and behaviours especially developed in humans, such as semantic processing, language, and Theory of Mind[69]. Also, it has been sustained that the orbitofrontal cortex may mediate reward-related behaviours, learning, decision making and expectation[70].

A remarkable finding of the present study is the specific HARs' polygenic association with SA rather than CT variability. This could be interpreted considering that the genetic influences on the two cortical measures and the underlying mechanisms are largely independent and follow distinct developmental trajectories[71–73], but both are related to neurodevelopmental genetic control[53,74,75]. As posited by the radial unit hypothesis, cortical SA expansion would be driven by the proliferation of neural progenitor cells, while CT would be determined by the number of their neurogenic divisions[76]. In this line, a study described that HARs-genes are highly expressed during prenatal development; their expression is upregulated during neurogenesis and enriched in cells from the outer radial glia[45]. Indeed, radial glia is a major class of neural stem cells in the germinal layer that shows substantial expansion in the primate lineage and, among the neurodevelopmental differences between humans and chimpanzees, there is the proliferative capacity of neural progenitors during cortical development[77]. Recent data shows that the genetic determinants of SA are predominately related to gene regulatory activity in neural progenitor cells during foetal development, while CT is influenced by regulatory processes that occur after mid-foetal development[53]. Also, common genetic variants explained a larger part of SA variance (SNP-$h^2$ = 34%, SE = 3%) than CT variance (SNP-$h^2$ = 26%, SE = 2%)[53]. However, our results could also be influenced by the reduced number of SNPs in our PRS estimates, which could have hampered the capture of the genetic determinants of CT. Likewise, by using HARs, we are focusing on regions highly stable along mammal evolution that have experienced rapid sequence changes in the human lineage since the divergence from our closest relatives. Moreover, while SA has enormously increased during the evolution of primates, cortical thickness has remained relatively constant[78]. Therefore, other genetic evolutionary markers could be more suitable for inspecting the evolutionary traces of CT.

Relative to the exploratory gene mapping results, it was interesting that the SNPs underlying cortical surface area differences within patients were enriched in biological processes essential for nervous system development. These findings align with previous studies describing that HARs-associated genes mainly participate in biological processes and pathways related to neurodevelopment, neural differentiation and axonogenesis[43].

Finally, we should account for some limitations of this study. Regarding our genetic association approach, the samples could be considered small; nonetheless, our study focused on the neuroimaging association approach. According to a recent revision, these analyses have been conducted in a larger sample than the median sample size of neuroimaging association studies to date[79]; however, it does not exclude the need for new studies in larger samples. In this regard also, we have to point out that our structural images were obtained from two scanners, which could represent a source of bias. Notwithstanding, we did not detect differences in neuroanatomical measures based on the two scanning sites, all the images passed the standardised quality-control protocols recommended by the ENIGMA consortium, which have been previously applied in large-scale multi-centre studies, and the scanner site was accounted as a covariate in the regressions. In terms of the genetic data, we should contemplate that our PRS estimates are pondered using SZ genetic burden, and the use of other GWAS summary statistics, such as the corresponding to the cortical phenotypes, could lead to different effects. The PRS estimation method used in the present study, PRS-C + T, is the most used, and the latest SZ GWAS has been conducted using the same method; however, other PRS calculation methodologies could be helpful[80]. Speaking of PRS estimations, our results are based on a sample of European ancestry and SZ GWAS statistics were derived from the European cohort. Then, although GWAS studies performed in non-European samples converge in the same SZ's genes and pathways[81], the extrapolation of our findings to other ethnic groups should not be done straightforwardly, and research based on populations of different ethnic origins should be encouraged. We should also consider that upcoming studies would greatly benefit from assessing environmental risk factors occurring along the neurodevelopmental period, which could modulate the genetic background and impact the brain developmental trajectories[82,83]. Finally, future study designs willing to understand the role of HARs in the neurobiological roots of SZ would benefit from analyses on other brain-based phenotypes such as structural connectivity, white matter microstructure[84–86], or MRI protocols related to social cognition[87].

Our findings, together with the increasing knowledge of the functions of HARs and the biological mechanisms in which they are involved, open new investigation venues. Based on the role of HARs as paramount actors in neurodevelopmental transcriptional regulation and their involvement in the genetic burden of neurodevelopmental psychiatric disorders like SZ and autism[42,43,47,51,65,88], our study and most of the previous research have been focused on HARs' common variability. Still, to gain insights into the pathogenic role of these regions, future studies should explore the functional effects of HARs' rare variability. Also, the prioritisation and interpretation strategies in whole-genome sequencing approaches should consider not only exonic or promoter variants but regulatory regions such as HARs. Furthermore, psychiatric disorders with a strong neurodevelopmental component are intimately related to the emergence of the human condition, sustained by the evolution of human-specific ontogenic mechanisms. Thus, studies on the genetic basis of this disorder should not be separated from this evolutionary component. Future studies would be strengthened by analysing the contribution of evolutionary relevant regions towards the disorders' genetic background[89–91].

In conclusion, our study adds evidence on the role of the genetic variability within HARs guiding foetal neurodevelopment and shaping cortical surface area configuration in patients with SZ. The biological plausibility of our findings highlights the importance of HARs in the early developmental gene regulatory machinery and suggests that these regions may contribute to bridging together the neurodevelopmental and evolutionary hypotheses in schizophrenia.

## Methods

**Sample**. The initial sample consisted of a case-control dataset of 378 individuals, of which 284 passed both the genetic and neuroimaging quality control (see details in the corresponding *Molecular analyses* and *MRI data acquisition* sections). Patients were recruited from inpatient and outpatient units at various centres from the Germanes Hospitalàries in Barcelona province. Healthy controls (HC) consisted of individuals from the same area, including non-medical staff employed at healthcare facilities, their relatives, and acquaintances, as well as members of the community recruited through online advertisements and independent sources. The patients' diagnoses were confirmed according to DSM-IV-TR based on an interview with two psychiatrists. Patients' symptoms were assessed based on the PANSS[92,93]. All participants were of European ancestry, between 18 and 65 years old, right-handed (based on self-report) and had an estimated intelligence quotient (IQ) (premorbid IQ in

patients) higher than 70, as assessed using the Word Accentuation Test[94]. All participants met the same exclusion criteria, which included suffering from major medical illness, conditions affecting cognitive or brain function, neurological conditions, history of head trauma with loss of consciousness and present or history of drug abuse or dependence. Additionally, for HC, exclusion criteria also included personal or family history of psychiatric service contact or treatment.

A group matching procedure was conducted to minimise the differences across diagnostic groups while maximising the sample size. Therefore, the analyses were conducted in a sample of 115 HC and 128 patients diagnosed with SZ with no age and sex differences between them (Table 2).

All subjects signed a written consent after being fully informed about the procedures and implications of the study, approved by the Germanes Hospitalàries Research Ethics Committee, and performed following its guidelines and in accord with the Declaration of Helsinki. All ethical regulations relevant to human research participants were followed.

**Molecular analyses**. Genomic DNA was extracted from buccal mucosa through cotton swabs or peripheral blood cells using Realpure Saliva or Blood kits (Durviz, S.L.U., Valencia, Spain).

A genome-wide genotyping was performed using the Infinium Global Screening Array-24 v1.0 (GSA) BeadChip (Illumina, Inc., San Diego, California, U.S) at the Spanish National Cancer Research Centre, in the Human Genotyping lab (CeGen-ISCIII), resulting in the genotyping of 730,059 SNPs. After quality control, a dataset of 447,035 SNPs with the following characteristics was obtained: Hardy-Weinberg equilibrium in patients and healthy controls, SNP call rate higher than 98% and minor allele frequency (MAF) higher than 0.005. Individuals with an SNP missingness higher than 2% were excluded. In addition, through a principal component analysis, those individuals found to be related or not of European ancestry were also excluded. Next, re-phasing and imputation were performed using, respectively, Eagle[95] and Minimac4[96] and the Haplotype Reference Consortium dataset (HRC version r1.1)[97] hosted on the Michigan Imputation Server[96] (https://imputationserver.sph.umich.edu/). A MAF value of >1% and an imputation quality of $R^2 > 0.3$ were required for the inclusion of the variants into further analyses. Finally, our final SNP dataset included 7,606,397 genetic markers.

**Polygenic risk scores estimation**. Using SZ GWAS 2022 summary statistics from the European subsample[10], we estimated four different PRS using PLINK 1.90 software[98] based on the PRS-C + T methodology[99], as in several recent GWAS studies[10,53,100]. This method is defined as the sum of allele counts, weighted by estimated effect sizes obtained from the GWAS, after two filtering steps: LD clumping (based on the European population from phase 3 of the 1000 Genomes Project reference panel) and $p$-value thresholding.

First, we calculated the whole-genome PRS (Global $PRS_{SZ}$). The LD filtering was conducted by including the most significant SNP from any pair showing an LD $r^2 > 0.015$ within 1000 kb windows, resulting in a set of informative linkage-disequilibrium independent markers (98,121 SNPs). Subsequently, for the $p$-value thresholding, we considered a range of thirteen $p$-value thresholds: $p < 5 \times 10^{-8}$, $p < 5 \times 10^{-7}$, $p < 5 \times 10^{-6}$, $p < 5 \times 10^{-5}$, $p < 5 \times 10^{-4}$, $p < 5 \times 10^{-3}$, $p < 0.05$, $p < 0.1$, $p < 0.2$, $p < 0.3$, $p < 0.4$, $p < 0.5$, $p < 1.0$. Through logistic regression, we established the best threshold in $p < 5 \times 10^{-3}$ as the better predictor of the diagnosis status based on Nagelkerke's pseudo-$R^2$ ($p = 4.75 \times 10^{-12}$, $R^2 = 0.22$).

Second, we estimated a PRS accounting for HARs genetic variability (HARs $PRS_{SZ}$), exclusively including SNPs within the 3070 autosomal HARs sequences compiled by Girskis et al.[47] (Supplementary Data 6). Using Bedtools 2.30.0[101], we selected the genetic markers in our sample within these HARs sequences. Considering the number of SNPs within the HARs, the PRS estimation was conducted following the same PRS-C + T methodology but adjusting LD clumping parameters. Following PLINK default options, we selected the most significant SNPs within 250 kb windows and with LD $r^2 > 0.5$ and the unique $p$-value threshold was set at $p < 1.0$. The final set of variants comprised 2201 SNPs (Supplementary Data 7).

Third, to assess the effect of HARs SNPs specifically affiliated with active foetal brain (FB) or adult brain (AB) gene regulatory elements, we estimated two additional PRS scores (FB-HARs $PRS_{SZ}$ and AB-HARs $PRS_{SZ}$) with the same procedure and parameters as for PRS-HARs. We followed the same methodology used in the latest ENIGMA human cerebral cortex GWAS[53] to do so. As described by Grasby et al., 2020[53], we downloaded ChromHMM chromatin states (Core 15-state model) from the Epigenomics Roadmap[102]. We selected the genomic regions comprising active regulatory elements (active transcription start site (TssA), flanking active transcription start site (TssAflnk), enhancers (Enh) and genic enhancers (EnhG)) for the two available foetal tissues (E081=foetal brain female and E082 = foetal brain male) and the four available cortical adult tissues (E067 = brain angular gyrus, E069 = brain cingulate gyrus, E072 = brain inferior temporal lobe and E073 = brain dorsolateral prefrontal cortex). We combined the foetal (E081 and E082)

**Table 2 Sample characteristics, including demographic and clinical description.**

|  | Healthy Controls | Patients with SZ |  |
| --- | --- | --- | --- |
| Age | 38.44 (11.98) | 40.15 (10.96) | $t$-test = −1.17, $p = 0.24$ |
| Sex | 60/55 (52.20%) | 82/46 (64.10%) | $\chi2 = 3.53$, $p = 0.06$ |
| Premorbid IQ | 103.52 (8.43) | 101.05 (9.16) | $t$-test = 2.18, $p = 0.03$ |
| Age at onset[a] | – | 21.00 (7.3) |  |
| Illness duration[a] | – | 17.30 (10.74) |  |
| PANSS Total[b] | – | 66.48 (18.65) |  |
| PANSS Positive[b] | – | 15.66 (5.88) |  |
| PANSS Negative[b] | – | 19.34 (7.31) |  |
| PANSS General Psychopathology[b] | – | 31.48 (9.15) |  |
| CPZ equivalents[c] | – | 581.28 (573.64) |  |

The quantitative variables include mean and standard deviation (sd). Sex description includes male/female count (% of males). Illness duration is given in years, and Chlorpromazine (CPZ) equivalents in mg/day.
[a]Age at onset and Illness duration (estimated by subtracting the age at onset to the current age) were available for 122 patients.
[b]PANSS scores were available for 124 patients.
[c]CPZ equivalent doses were available for 126 patients.

and adult (E067, E069, E072 and E073) annotations and selected only those regions non-overlapping between them as foetal brain-specific and adult brain-specific. With the selected HARs PRS$_{SZ}$ SNPs, we selected the genetic variants allocated within these foetal and adult brain-specific regions. The final set of variants included in the FB-HARs PRS$_{SZ}$ and AB-HARs PRS$_{SZ}$ estimations were 112 and 81 SNPs, respectively (Supplementary Data 7), and the data on the four estimated PRSs can be found at Supplementary Data 8.

**MRI data acquisition**. The MRI neuroimaging data were obtained from two scanners: 58% (70 HC, 72 patients) of the sample was scanned in a 1.5 T GE Sigma scanner (General Electrical Medical Systems, Milwaukee, Wisconsin, USA) and the other 42% (45 HC, 56 patients) in a 3 T Philips Ingenia scanner (Philips Medical Systems, Best, The Netherlands) at Hospital Sant Joan de Déu (Barcelona, Spain).

High-resolution structural-T1 MRI data in the 1.5 T scanner was obtained using the following acquisition parameters: matrix size $512 \times 512$; 180 contiguous axial slices; voxel resolution $0.47 \times 0.47 \times 1$ mm$^3$; echo time (TE) = 3.93 ms, repetition time (TR) = 2000 ms; and flip angle = 15°. At the 3 T scanner, structural T1-weighted sequences were acquired as follows: matrix size $320 \times 320 \times 250$; voxel resolution $0.75 \times 0.75 \times 0.80$ mm$^3$; TE = 3.80 ms, TR = 8.40 ms; and flip angle = 8°. All images were visually inspected to exclude those with artefacts and movement.

**Neuroanatomical data**. Structural MRI data were processed using the FreeSurfer image analysis suite (http://surfer.nmr.mgh.harvard.edu/). The images obtained from the two different scanners were independently pre-processed. This process included the removal of non-brain tissue, an automated Talairach transformation, tessellation of the grey and white matter boundaries and surface deformation[103], after which individual images were normalised to a common stereotaxic space. Several deformation procedures were performed in the data analysis pipeline, including surface inflation and registration to a spherical atlas. This method uses both intensity and continuity information from the entire three-dimensional images in the segmentation and deformation procedures to produce vertex-wise representations of CT and SA. The CT was defined as the measure of the distance between the white matter surface and the pial surface, and cortical SA was calculated as the area of the white matter surface. With FreeSurfer, we automatically performed the segmentation of 34 cortical regions of interest for each hemisphere using the Desikan-Killiany cortical atlas[104]. Within these defined regions, mean values of CT and SA were quantified for each individual. Additionally, the global mean CT and the total SA were also considered.

All subjects included in the study passed the standardised quality-control protocols from the ENIGMA consortium (https://enigma.ini.usc.edu/protocols/imaging-protocols/) previously applied in large-scale multi-centre studies[53,105].

**Statistics**. Demographic and clinical data were processed and analysed using SPSS (IBM SPSS Statistics, version 29.0, released 2022, IBM Corporation, Armonk, New York, USA).

We compared the SZ polygenic load of patients and controls based on the four different PRS estimations (Global PRS$_{SZ}$, HARs PRS$_{SZ}$, FB-HARs PRS$_{SZ}$ and AB-HARs PRS$_{SZ}$) by means of block-wise logistic regressions with the two diagnostic groups (SPSS). For each PRS, two statistical models with case/control status as outcome were compared, one testing the covariates alone (age, sex and the two first ancestry-specific principal components

as a baseline model) and the other testing the covariates plus the corresponding PRS (full model). We report the $R^2$ values as the differences in Nagelkerke's pseudo-$R^2$ between these two nested models as an indicator of explained variance[60,106]. To assess whether PRSs modulate patients' clinical profiles, we tested the effect of the four PRSs estimations on age at onset, illness duration, PANSS scores, and antipsychotic medication dose by means of linear regression models (controlled by sex).

Next, we examined to which extent the four different PRSs modulate CT and SA measures of each cortical region. We applied linear regression models, separately in HC and patients, to test the effect of each of the four PRS (as independent variables) on both neuroanatomical measures (as dependent variables) (R software). To assess whether the diagnostic status modulates the PRS effect, we conducted linear models using the whole sample and tested the PRS x diagnosis interaction. To control for the differences in scanners among individuals plus the potential effects of sex, age, premorbid IQ, and intracranial volume, all these variables were included as covariates in the analyses. In the linear regression within patients with SZ, the antipsychotic dose was also included as a covariate[19,107].

According to the findings, to determine if the detected anatomical changes were related to the clinical profiles of patients, we analysed whether the SA changes impacted their age at onset, illness duration, and PANSS scores by means of linear regression models. To do this, we regressed the SA residuals from the four significant cortical regions of interest (obtained after regressing age, sex, intracranial volume, scanner, and antipsychotic medication) on the clinical phenotypes (controlling for age, sex, and antipsychotic medication) (SPSS).

Lastly, the SNPs included in the FB-HARs PRS$_{SZ}$ were furtherly mapped and functionally annotated using FUMA[108] with the SNP2GENE and the GENE2FUNC tools. The positional mapping parameters were left as default. The eQTL mapping was conducted in PsychENCODE, ComminMind and BRAINEAC tissues filtering by PsychENCODE and brain open chromatin atlas annotations. The 3D chromatin interaction mapping was built-in PsychENCODE and Hi-C adult and foetal cortex, dorsolateral and hippocampus and neural progenitor cells data, filtered by PsychENCODE and brain open chromatin atlas annotations.

The p-values from each of the before mentioned statistical tests were adjusted using the false discovery rate (FDR) method, precisely the Benjamini-Hochberg procedure, to control for multiple comparisons at level $q = 0.05$. Accordingly, only those results with a corrected FDR-pval<0.05 are reported as statistically significant.

The significant results on the cortical regions were plotted using the *ggseg* library in R[109] and the regressions with the direction of the results were plotted using SPSS (the numerical source data for graphs is provided in Supplementary Data 9).

**Reporting summary**. Further information on research design is available in the Nature Portfolio Reporting Summary linked to this article.

## Data availability
The data supporting the findings of this study are available from the corresponding authors upon reasonable request.

## Code availability
The custom code used for the analyses of this study is available from the corresponding authors upon reasonable request.

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

## Acknowledgements

This study received project funding from the Brain & Behavior Research Foundation (NARSAD Young Investigator Award) to MF-V (grant ID 26206). Researchers were supported by: (i) Instituto de Salud Carlos III (ISCIII) through the contracts FI19/0352 to MG-R, CD22/00106 to MAG-L, and CP20/00072 to MF-V (co-funded by European Regional Development Fund (ERDF)/European Social Fund "Investing in your future"); (ii) la "Caixa" Foundation through the Junior Leader Fellowship contract LCF/BQ/PR22/11920017 to PF-C; (iii) the Swiss National Science Ambizione grant PZ00P2_185814 to EJC-R; (iv) Júlia Gil Pineda Research Fellowship to ER-C. We thank the Comissionat per a Universitats i Recerca del DIUE of the Generalitat de Catalunya (Agència de Gestio d'Ajuts Universitaris i de Recerca (AGAUR, 2021SGR01475). CeGen PRB3 is supported by grant PT17/0019 of the PE I + D + i 2013–2016, funded by ISCIII and ERDF.

## Author contributions

M.G.-R. and M.F.-V. conceived the study. M.G.-R. conducted the DNA extraction and sample normalisation for genotyping. E.J.C.-R., M.A.G.-L. and P.F.-C. pre-processed and segmented the MRI images. E.P.-C., P.F.-C., J.S., J.T., L.T. and E.R.-C. conducted the recruitment and/or the clinical evaluation. E.J.C.-R., R.S. and E.P.-C. designed the MRI protocols. M.G.-R., C.A. and A.A. performed the data curation. M.G.-R. conducted the formal statistical analyses and graphical representations with the help of C.A.-P., A.A., and M.L.-G. S.P. and M.F.-V. guided and reviewed the methodology. M.G.-R. and M.F.-V. interpreted the results. M.G.-R. and M.F.-V. wrote the first and subsequent drafts of the paper. M.F.-V. supervised the study activity planning and execution. M.F.-V. and E.P.-C. participated in the funding acquisition. All the authors reviewed and approved the final manuscript.

## Competing interests

The authors declare no competing interests.
