## [Peer Review File · Communications Biology]

Reviewers' comments:

Reviewer #1 (Remarks to the Author):

The authors found new findings by combining genetic and with brain imaging analyzes such as HARs' polygenic load and polygenic risk scores in patients with schizophrenia. It is particularly interesting to show the HAR associated with the fetal brain and the relevant brain regions of schizophrenia, however there are several concerns against the result and discussion they showed. The comments on the manuscript are as follows;

1. Age and gender are taken into account as covariates, but is it possible to ignore height and weight?
2. It is of great interest to what extent the authors' findings are related to symptoms of schizophrenia. Is it possible to do such an analysis from a diagnostic scale or something?
3. Likewise, in the discussion, it is important to discuss the authors' findings for a broader audience of associations between the superior temporal cortex, lateral orbitofrontal cortex, pars triangularis, paracentral lobules and symptoms of schizophrenia. I especially expect neuroanatomical and evolutionary discussions.
4. The figure in Supplementary Table 3 is easier for readers to understand if it is shown in the paper as the main figure.
5. I could not understand the reason why FB-HARs PRS and SA were analyzed. Add an explanation to the results section. Also, have you analyzed the correlation between AB-HARs PRS and SA?
6. The HAR-derived PRS showed no differences between groups, raising questions about the relevance of HAR to the description in the authors' abstract introduction. We need to argue this point explicitly and provide evidence for the evolutionary significance of the authors' findings.

Reviewer #2 (Remarks to the Author):

The authors investigate the influence of genetic variants in the HARs in modifying the neurodevelopmental trajectory in control subjects and patients diagnosed with schizophrenia using a polygenic score approach. They found that foetal brain HARs PRS was associated with significantly lower cortical surface area in the lateral orbitofrontal cortex, the superior temporal cortex, the pars triangularis and the paracentral lobule.

There is ample information on the variation in brain volumes in schizophrenia, and the association or overlap between genes in HARs and schizophrenia is also known. The authors address a gap on whether the schizophrenia related variants affect the brain cortical surface area. Although the authors raise an interesting question, there is room for a lot of improvement.

The manuscript needs more information on the patients. As schizophrenia is a condition that has varied manifestations in individuals, more information on the patient symptom severity would be useful. Following this, they could explore if the polygenic score is higher in individuals with severe symptoms or not. Or if the association is dependent on individuals with severe symptoms.

The age of onset of the condition would be important covariate. If not included in the analysis, at least they need to provide information on how much variation was there for the age of onset among patients.

The readers need to see the table of regression results with the effect sizes and p values shown for all the models tested. In addition to the standardized beta, the raw beta values also need to be included in the model. Since sex is an important variable, it would be informative to see if there was a group by gender interaction effect.

Parental SES measures or childhood SES measures have been known to influence the neurodevelopmental trajectories in childhood. The authors did not mention that in the study.

Why was total surface area or mean cortical thickness not considered in the model to show regional specificity.

The final set of variants included in the FB-HARs PRSSZ and AB-HARs PRSSZ was not given in the paper. Was the selection of foetal and adult tissues for selecting FB and AB specific HARs random or

any criteria was considered?

Reviewer #3 (Remarks to the Author):

This submission describes a case-control study demonstrating a modulatory effect of fetal active regulatory Human Accelerated Regions (HARs) on cortical measures of different brain regions in Schizophrenia. The study is generally strong and would be a meaningful contribution to the literature. However, there are several areas upon which the writing could be improved, including shortening and simplifying the Introduction. Several run-on sentences are present and serve to weaken the overall flow. Additionally, this section could be better organized to introduce a stepwise argument for the importance of, and rationale for, this research in the context of its findings. Please find below more specific suggestions by section that may further strengthen the manuscript:

INTRODUCTION

- 1) Paragraph 1, Line 32 – does social cognition distinguish humans from other species, or a particular aspect/approach to social cognition?
- 2) Paragraph 2, Line 51 – “latter” should be later? This could also be removed altogether.
- 3) Paragraph 5, Line 81 – default mode networks is written out without an acronym in parenthetical notation. The subsequent sentence uses DMN and then authors returns to the full term. Given that “default mode network” seems to only be used a total of three times, I recommend removing the acronym altogether.
- 4) Line 98 – PRS is mentioned for the first time here; more references to the extant literature should be used in this section.

METHODS

Sample

- 5) Paragraph 1, Line 114 - it is stated that “healthy controls were recruited from the same area.” This refers to Barcelona, Spain? How were HCs recruited and were they compensated?
- 6) Paragraph 1, Line 115 – “patients’ diagnosis” should be diagnoses.
- 7) Paragraph 1, Line 116 - were all individuals right-handed without reported ambidexterity or nuances? If not, how was laterality determined?
- 8) Paragraph 1, Line 119 – did this study take place during the COVID-19 pandemic and, if so, was previous infection considered an exclusionary condition “affecting cognitive or brain function”?
- 9) Table 1 – would median be a more appropriate value for illness duration?

Molecular Analyses

- 10) Paragraph 2, Line 140 – this is the first mention of SNPs (no literature mentioned in Introduction); though generally understood, this should also be spelled out with acronym separate at first mention.
- 11) Paragraph 2, Line 144 – it is unnecessary to provide an acronym (e.g., PCA) if the term is used ≤ 2 times.
- 12) Paragraph 2, Line 144 – how many individuals were excluded due to not being of European ancestry and why was this decision made? This will serve as a limitation.

Surface-Based Morphometry

- 13) Paragraph 1, Line 195 – in the context of FreeSurfer, how did the authors control for the differences in scanners among individuals (e.g., 1.5T vs. 3T; GE vs. Phillips)? This is noted as a limitation but should be noted in the relevant section that FreeSurfer does not handle this.

RESULTS

PRS Associations with Morphometric Measures

- 14) Paragraph 3, Line 266 – see Item 11 regarding “Gene Ontology (GO)”.

DISCUSSION

- 15) Paragraph 1, Line 275 – why do the authors believe that the effect of HARS genetic variability on cortical measures in SZ vs. HC has not been studied previously?
- 16) Paragraph 4, Line 309 – remain consistent with use of SZ.
- 17) Paragraph 4, Line 313 – use superscript citation for Wei et al. (2019).
- 18) I recommend discussing future directions stemming from this research.

Reviewer #1 (Remarks to the Author):

The authors found new findings by combining genetic and with brain imaging analyzes such as HARs' polygenic load and polygenic risk scores in patients with schizophrenia. It is particularly interesting to show the HAR associated with the fetal brain and the relevant brain regions of schizophrenia, however there are several concerns against the result and discussion they showed. The comments on the manuscript are as follows;

1. Age and gender are taken into account as covariates, but is it possible to ignore height and weight?

We agree with the reviewer regarding the importance of controlling for anthropometric measurements in this type of analyses assessing brain cortical architecture. This is why we covariate all the regressions modelling the cortical thickness and the surface area variability for the intracranial volume. Additionally, during the MRI images pre-processing there is an automated Talairach transformation, tessellation of the grey and white matter boundaries and surface deformation procedure, which normalises individual cortical maps. We have clarified these methodological aspects in the corresponding methods sections.

Methods, Surface-based morphometry section, page 8-9, lines 198-201: "Structural MRI data were processed using the FreeSurfer image analysis suite (<http://surfer.nmr.mgh.harvard.edu/>). The images obtained from the two different scanners were independently pre-processed. This process included the removal of non-brain tissue, an automated Talairach transformation, tessellation of the grey and white matter boundaries and surface deformation⁶⁸, after which individual images were normalised to a common stereotaxic space."

Methods, Statistical analyses section, page 10, lines 229-233: "In order to control for the differences in scanners among individuals plus the potential effects of sex, age, premorbid IQ and intracranial volume, all these variables were included as covariates in the analyses. In the linear regression within patients with SZ, the antipsychotic dose was as well included as a covariate^{19,73}."

2. It is of great interest to what extent the authors' findings are related to symptoms of schizophrenia. Is it possible to do such an analysis from a diagnostic scale or something?

We agree with the reviewer about the interest of assessing patients' symptoms in relation to the genetic load.

Therefore, we included in the analyses the assessment of patients' symptoms based on the Positive and Negative Syndrome Scale (PANSS) in addition to other clinical variables of interest such as age at onset, illness duration and medication dose. The clinical profile of the patients has been included in the table describing the sample (Table 1).

Also, we conducted several analyses. On the one hand, we tested the effect of the four PRSs on PANSS scores and the other clinical variables, by means of linear regressions (covaried by sex). The results revealed that none of the four PRSs significantly modulated patients' symptomatology.

On the other hand, we tested whether the surface area in the regions where we detected the PRS HARs-FB effect was associated with the patients' clinical profile. We conducted linear regressions (using the SA residuals, controlled for age, sex, premorbid IQ, intracranial volume, scanner and medication, as independent variables) and the PANSS scores, age at onset, illness duration, and medication (as dependent variables). These models were adjusted by age, sex and medication, (except for the medication regressions). There were no significant results after the FDR correction.

These additional analyses and the findings have been included in the manuscript.

Methods, Statistical analyses section, page 10, lines 222-224: “To assess whether PRSs modulate patients’ clinical profile, we tested the effect of the four PRSs estimations on age at onset, illness duration, PANSS scores, and medication dose by means of linear regressions models (controlled by sex).”

Methods, Statistical analyses section, page 10, lines 233-237: “According to the findings, to determine if the detected anatomical changes were related to the clinical profile of patients, we analysed whether the SA changes impacted their age at onset, illness duration, and PANSS scores by means of linear regression models. To do this, we regressed the SA residuals from the four significant cortical regions of interest (obtained after regressing age, sex, intracranial volume, scanner and medication), on the clinical phenotypes (controlling for age, sex and medication).”

Results, PRS comparison between diagnostic groups section, page 10, lines 253-256: “Diagnostic PRS comparisons revealed differences in the Global PRS_{SZ}, with patients presenting higher SZ genetic load than HC. The HARs-derived PRSs did not show between-groups differences (**Table 2**). No significant PRSs effect on age at onset, illness duration, PANSS scores, or medication was detected (*Supplementary Table 3*).”

Results, PRS associations with morphometric measures section, page 11, lines 276-278: “The SA variability in these four regions of interest was not related to the patients’ clinical profile (*Supplementary Table 5*).”

3. Likewise, in the discussion, it is important to discuss the authors' findings for a broader audience of associations between the superior temporal cortex, lateral orbitofrontal cortex, pars triangularis, paracentral lobules and symptoms of schizophrenia. I especially expect neuroanatomical and evolutionary discussions.

We agree with the reviewer on the importance of exploring and discussing the structural findings in relation to patients’ symptoms. Thus, we have explored whether the changes in SA among patients were associated with their clinical profile in terms of the severity of positive symptoms, negative symptoms, and general psychopathology (as assessed by the PANSS scale), the age at onset, and illness duration. These additional analyses revealed no relationship between SA and patients’ clinical profile. These finding, aligns with previous results from a larger meta-analysis on cortical architecture in schizophrenia (van Erp et al., *Biological Psychiatry* 2018).

Relative to the discussion of the structural findings from the point of view of schizophrenia symptoms and brain evolution, we have added several points in the corresponding sections.

Discussion, page 14, lines 330-337: “In spite of the detected differences in SZ, the reductions in SA were not associated with the patients’ symptoms, onset of the disorder or its duration. These outcomes converge with the findings from van Erp et al., who reported that the regional SA changes in patients were not correlated with PANSS total, positive, and negative symptom scores¹⁹. Even so, SA reductions in the right frontal cortex have been correlated with increased negative symptoms in patients with psychosis including SZ, schizoaffective disorder, and bipolar disorder type I with psychosis⁸³. In the same line, other structural and functional alterations in the superior temporal region have been reported in relation to formal thought disorder in patients^{84,85}.”

Discussion, page 14-15, lines 344-348: “Moreover, it has been proposed that the anatomical changes in the human temporal cortex compared to other primates may be involved in higher-level cognitive functions and behaviours especially developed in humans, such as semantic processing, language, and theory of mind⁸⁶. Also, it has been sustained that the orbitofrontal cortex may mediate reward-related behaviours, learning, decision making and expectation⁸⁷.”

4. The figure in Supplementary Table 3 is easier for readers to understand if it is shown in the paper as the main figure.

According to the suggestion, we have included the figures that were in Supplementary Table 3 in the manuscript main body as Figure 2.

Results, PRS associations with morphometric measures section, page 12: **“Figure 2. Gene set enrichment results. Functional annotation results in Gene Ontology (GO) categories for biological processes (upper panel) and cellular components (lower panel). In each panel, the GO categories can be seen on the left, followed by the proportion of the overlapping genes and the associated p-value (FDR corrected), and the overlapping genes on the right.”**

5. I could not understand the reason why FB-HARs PRS and SA were analyzed. Add an explanation to the results section. Also, have you analyzed the correlation between AB-HARs PRS and SA?

We apologise if the methods description was misleading. We have assessed the four different polygenic risk scores (PRSs) on both cortical anatomical measures, the surface area (SA) and the cortical thickness (CT). To clarify this, we have re-formulated the statistical analysis paragraph in the methods section.

Methods, Statistical analyses section, page 9, lines 225-227: **“Next, we examined to which extent the four different PRSs modulate CT and SA measures of each cortical region. We applied linear regression models (R software), separately in HC and patients, to test the effect of each of the four PRS (as independent variables) on both neuroanatomical measures (as dependent variables).”**

6. The HAR-derived PRS showed no differences between groups, raising questions about the relevance of HAR to the description in the authors' abstract introduction. We need to argue this point explicitly and provide evidence for the evolutionary significance of the authors' findings.

In the abstract introduction our intention was to highlight that previous studies have associated HARs variability with diagnosis (see Xu et al. 2015, Srinivasan et al. 2017). In accordance with the reviewer’s suggestion, we have restructured the abstract to emphasise that the association between HARs and schizophrenia diagnosis was made in previous studies, not in ours. As well, we have included the negative findings on HARs PRSs effect on the risk in the abstract to avoid confusion.

Abstract, page 2, lines 13-25: **“Schizophrenia may represent a trade-off in the evolution of human-specific neurodevelopmental ontogenetic mechanisms. Human Accelerated Regions (HARs) are evolutionary markers functioning as neurodevelopmental transcription enhancers that have been previously associated with brain configuration, neural information processing, and schizophrenia risk. We investigated the influence of HARs’ polygenic load on neuroanatomical measures through a case-control approach (128 patients with schizophrenia and 115 controls). To this end, we calculated the global schizophrenia polygenic burden (Global PRS_{SZ}) and that specific to HARs (HARs PRS_{SZ}), as well as those specific to HARs linked to foetal brain (FB-HARs PRS_{SZ}) and adult brain (AB-HARs PRS_{SZ}) transcriptional regulatory elements. We explored the PRSs’ main effects and PRSs x diagnosis interactions on regional cortical thickness (CT) and surface area (SA). Results indicate that higher FB-HARs PRS_{SZ} is associated with patients’ lower SA in the lateral orbitofrontal cortex, the superior temporal cortex, the pars triangularis and the paracentral lobule. While none of the HARs-derived PRSs showed an effect on the risk, the neuroanatomical findings suggest an involvement of the human-**

specific prenatal neurodevelopmental transcriptional regulation in SA, highlighting the role of evolutionary markers in the schizophrenia genomic architecture.”

Also, regarding reviewer’s suggestion to argue the undetected diagnostic differences with HARs-derived PRSs and diagnostic groups from an evolutionary perspective, we have included several additional points in the corresponding discussion section.

In this sense, we highlighted that, to our knowledge, no study has previously inspected HARs effect on schizophrenia risk using PRSs methods, and therefore, our findings should be replicated in larger samples to further investigate the direct association of HARs polygenic load with schizophrenia risk. Still, based on the increasing number of data that do support the association of HARs sequence variability with SZ, from both candidate and genome-wide approaches, the importance of these evolutionary regions in the disorder’s genetic background should not be directly dismissed.

Discussion, pages 13, lines 304-310: “However, in our sample, the analyses indicate no significant differences across diagnostic groups using the HARs-related PRS_{SZ}. The lack of association between HARs-derived PRSs and schizophrenia risk in our sample may be indicative of a broader genotypic background underlying the diagnosis, but it could also result from insufficient power. Therefore, future association studies with larger case-control samples are needed to confirm, or dismiss, the direct link between HARs polygenic load and the risk of developing SZ. Whereas the direct comparison of our findings with previous research is difficult because of the absence of HARs-based PRS studies, some previous SZ association studies exploring the variability in candidate HARs point towards a link between these evolutionary regions and the risk for the disorder.”

Reviewer #2 (Remarks to the Author):

The authors investigate the influence of genetic variants in the HARs in modifying the neurodevelopmental trajectory in control subjects and patients diagnosed with schizophrenia using a polygenic score approach. They found that foetal brain HARs PRS was associated with significantly lower cortical surface area in the lateral orbitofrontal cortex, the superior temporal cortex, the pars triangularis and the paracentral lobule.

There is ample information on the variation in brain volumes in schizophrenia, and the association or overlap between genes in HARs and schizophrenia is also known. The authors address a gap on whether the schizophrenia related variants affect the brain cortical surface area. Although the authors raise an interesting question, there is room for a lot of improvement.

The manuscript needs more information on the patients. As schizophrenia is a condition that has varied manifestations in individuals, more information on the patient symptom severity would be useful. Following this, they could explore if the polygenic score is higher in individuals with severe symptoms or not. Or if the association is dependent on individuals with severe symptoms.

We agree with the reviewer that it is important to characterise patient's clinical profile considering the heterogeneous manifestation of the disorder. In this sense, at the moment of the MRI session, all patients were clinically stable according to their psychiatrists' assessment. Additionally, at the moment of the MRI patients' symptoms were assessed with the Positive and Negative Syndrome Scale (PANSS). According to the reviewer's suggestions we have included the information on the PANSS total, positive, negative and general psychopathology scores in the table describing the sample (Table 1).

Moreover, we have tested whether the clinical profile of the patients (age at onset, illness duration, PANSS scores, and medication) was influenced by any of the PRSs used (by means of linear regressions covaried by sex). None of the four PRSs significantly modulated patients' clinical profile. Similarly, we tested whether the SA from the cortical regions significantly modulated by the PRS_{SZ} HARs-FB influenced these clinical variables. We conducted linear regressions (using the SA residuals, controlled for age, sex, premorbid IQ, intracranial volume, scanner and medication, as independent variables) and the PANSS scores, age at onset, illness duration, and medication (as dependent variables, covaried by age, sex and medication, except for the medication regressions). There were no significant results after the FDR correction.

We have included these analyses in the manuscript and the description of the results can be found in the main body of the text as well as in the corresponding supplementary tables.

Methods, Statistical analyses section, page 10, lines 222-224: *"To assess whether PRSs modulate patients' clinical profile, we tested the effect of the four PRSs estimations on age at onset, illness duration, PANSS scores, and medication dose by means of linear regressions models (controlled by sex)."*

Methods, Statistical analyses section, page 10, lines 233-237: *"According to the findings, to determine if the detected anatomical changes were related to the clinical profile of patients, we analysed whether the SA changes impacted their age at onset, illness duration, and PANSS scores by means of linear regression models. To do this, we regressed the SA residuals from the four significant cortical regions of interest (obtained after regressing age, sex, intracranial volume, scanner and medication), on the clinical phenotypes (controlling for age, sex and medication)."*

Results, PRS comparison between diagnostic groups section, page 10, lines 253-256: “Diagnostic PRS comparisons revealed differences in the Global PRS_{SZ}, with patients presenting higher SZ genetic load than HC. The HARs-derived PRSs did not show between-groups differences (**Table 2**). *No significant PRSs effect on age at onset, illness duration, PANSS scores, or medication was detected (Supplementary Table 3).*”

Results, PRS associations with morphometric measures section, page 11, lines 276-277: “*The SA variability in these four regions of interest was not related to the patients’ clinical profile (Supplementary Table 5).*”

The age of onset of the condition would be important covariate. If not included in the analysis, at least they need to provide information on how much variation was there for the age of onset among patients.

We agree with the reviewer that age at onset is an important variable for describing patients’ course of illness as it is related to symptoms and with higher genetic burden. We have included this variable in the table describing the sample of the study (Table 1).

As suggested by the reviewer, we also verified that there were no differences between patients with an early onset (before 18 y) and those with adult onset (after 18 y) in relation to the genetic burden in none of the estimated PRSs (T-test findings).

We discussed the suitability of using age at onset or illness duration as covariates in the analyses within patients. Nonetheless, we ruled out this idea for two main reasons. First, because both measures significantly correlate with age (age at onset $r=0.35$, $p<0.001$; illness duration $r=0.80$, $p<0.001$) and including them in the model would raise collinearity issues. Second, because based on our aim of testing the effect of the four PRSs on cortical measures within controls, patients, and the PRSs x diagnosis interaction, age was the common covariable across all analyses that allowed the comparability of the findings.

The readers need to see the table of regression results with the effect sizes and p values shown for all the models tested. In addition to the standardized beta, the raw beta values also need to be included in the model. Since sex is an important variable, it would be informative to see if there was a group by gender interaction effect.

Considering the reviewer’s recommendation, we have included in supplementary tables (see Supplementary Table 4) the regression results for all the tested models. Additionally, the raw beta value of the significant findings have been also included in the description of the results in the main body of the manuscript.

Results, PRS associations with morphometric measures section, page 11, lines 267-273: “In contrast, the linear regression analyses revealed that among patients with SZ, the FB-HARs PRS_{SZ} significantly affected cortical SA in different regions of the right hemisphere. FB-HARs PRS_{SZ} modulated the patients’ SA of the lateral orbitofrontal cortex ($\beta=-1581.796$, Standardised $\beta=-0.234$, $SE=440.443$, adjusted $R^2=0.491$, $FDR-pval=0.008$), the superior temporal cortex ($\beta=-1960.151$, Standardised $\beta=-0.235$, $SE=545.898$, adjusted $R^2=0.488$, $FDR-pval=0.008$), the pars triangularis ($\beta=-1409.886$, Standardised $\beta=-0.242$, $SE=438.928$, adjusted $R^2=0.322$, $FDR-pval=0.020$) and the paracentral lobule ($\beta=-842.876$, Standardised $\beta=-0.233$, $SE=282.910$, adjusted $R^2=0.264$, $FDR-pval=0.031$) (**Figure 1**).”

As well, following the second comment, we have repeated the regression analyses on the morphometrical measures to see if there was an interaction between the four estimated PRSs and sex on cortical thickness and surface area. Nonetheless, none of the PRSs x sex interactions was significant after FDR correction. Considering that these analyses were not the main focus of our investigation, we have considered not to

include these additional analyses/results in the manuscript nor in the supplementary information. But we will be happy to share the results with the reviewer if he/she considers it appropriate.

Parental SES measures or childhood SES measures have been known to influence the neurodevelopmental trajectories in childhood. The authors did not mention that in the study.

We totally agree with the review in the fact that environmental risk factors occurring along the course of the neurodevelopment would interact with the genetic background to modulate the brain developmental trajectories. Unfortunately, in our sample the conditions under which this neurodevelopment has occurred are unknown. Future studies would greatly benefit from characterising the neurodevelopment including information on obstetric complications occurring during pregnancy or in the perinatal period, childhood adversities such as the familial socioeconomical status, childhood trauma, as well as other factors such as social isolation, substance abuse, which are known environmental factors influencing the vulnerability towards schizophrenia and other neurodevelopmental disorders. We have added a comment on the necessity to better characterise the neurodevelopment in the limitations section.

Discussion, page 16, lines 392-394: “We should also consider that upcoming studies would greatly benefit from assessing environmental risk factors occurring along the neurodevelopmental period, which could modulate the genetic background and impact the brain developmental trajectories^{99,100}.”

Why was total surface area or mean cortical thickness not considered in the model to show regional specificity.

We apologise with the reviewer for forgetting to include this information in the manuscript. We indeed estimated the mean cortical thickness and the total surface area for each individual, and thus conducted the PRSs linear regressions with these measures as well. No significant PRSs effects were observed in relation to mean CT or total SA neither within controls, patients, nor in interaction with diagnosis. We have now incorporated this information into the manuscript and the detail of the results are accessible on supplementary tables.

Methods, Surface-based morphometry section, page 9, lines 207-209: “Within these defined regions, mean values of CT and SA were quantified for each individual. Additionally, the global mean CT and the total SA were also considered.”

The final set of variants included in the FB-HARs PRSSZ and AB-HARs PRSSZ was not given in the paper. Was the selection of foetal and adult tissues for selecting FB and AB specific HARs random or any criteria was considered?

According to the reviewer’s proposal, we have included a supplementary table with the final set of variants (with #rs) included for the PRSs estimation of the HARs PRS_{SZ}, FB-HARs PRS_{SZ} and AB-HARs PRS_{SZ} (see Supplementary Table 2). The reference to the supplementary table has been included in the main manuscript text.

Methods, Polygenic Risk Score Estimation section, page 8, lines 182-183: “The final set of variants included in the FB-HARs PRS_{SZ} and AB-HARs PRS_{SZ} estimations were 112 and 81 SNPs, respectively (*Supplementary Table 2*).”

The findings described in the latest ENIGMA Consortium GWAS on cortical anatomy describe different genomic signatures underlying surface area and cortical thickness variability (Grasby et al., 2020). Then, we followed the same criteria to assess the specific effects of HARs polygenicity associated with foetal brain and adult brain. In this sense, to assess the partitioned heritability explained by foetal and adult annotated genome regions we used the same procedure described in the ENIGMA paper (details can be which can be found at Grasby et al., 2020 supplementary materials).

We selected from the Epigenomics Roadmap data, the ChromHMM chromatin states comprising active regulatory elements (active Transcription start site (TssA), flanking active transcription start site (TssAflnk), enhancers (Enh) and genic enhancers (EngG)) for the two available foetal tissues (E081=foetal brain female and E082=foetal brain male) and the four available cortical adult tissues (E067=brain angular gyrus, E069=brain cingulate gyrus, E072=brain inferior temporal lobe and E073=brain dorsolateral prefrontal cortex).

We have clarified this selection criterion in the corresponding section to ensure that the reader is not misled.

Methods, Polygenic Risk Score Estimation section, page 8, lines 170-179: “Third, to assess the effect of HARs SNPs specifically affiliated with foetal brain (FB) or adult brain (AB) gene regulatory elements, we estimated two additional PRS scores (FB-HARs PRS_{SZ} and AB-HARs PRS_{SZ}) with the same procedure and parameters as for PRS-HARs. To do so, we followed the same methodology used in the latest ENIGMA human cerebral cortex GWAS⁵³. As described by Grasby et al., 2020⁵³, we downloaded ChromHMM chromatin states (core 15 state model) from the Epigenomics Roadmap⁶⁷ and selected the genomic regions comprising active regulatory elements (active transcription start site (TssA), flanking active transcription start site (TssAflnk), enhancers (Enh) and genic enhancers (EngG)) for the two available foetal tissues (E081=foetal brain female and E082=foetal brain male) and the available cortical adult tissues (E067=brain angular gyrus, E069=brain cingulate gyrus, E072=brain inferior temporal lobe and E073=brain dorsolateral prefrontal cortex).”

Reviewer #3 (Remarks to the Author):

This submission describes a case-control study demonstrating a modulatory effect of fetal active regulatory Human Accelerated Regions (HARs) on cortical measures of different brain regions in Schizophrenia. The study is generally strong and would be a meaningful contribution to the literature. However, there are several areas upon which the writing could be improved, including shortening and simplifying the Introduction. Several run-on sentences are present and serve to weaken the overall flow. Additionally, this section could be better organized to introduce a stepwise argument for the importance of, and rationale for, this research in the context of its findings. Please find below more specific suggestions by section that may further strengthen the manuscript:

We thank the reviewer for his/her comments. We have specifically assessed each of the points proposed, and additionally we have simplified the introduction to provide the reader with a more focused framework.

Introduction, page 2, lines 37-50: “Such pathways are tightly orchestrated and involve gene expression regulatory mechanisms of paramount importance for brain ontogeny ¹¹, which, at the same time, is highly influenced by the environmental events occurring during prenatal and early periods. Remarkably, prenatal and obstetric complications are associated with an increased risk for SZ ¹², and their presence correlates with a higher genomic risk for the disorder ¹³. In parallel, the placenta-associated genomic burden for schizophrenia is linked to reduced brain volumes in neonates and poorer cognitive development during the first two years of life ¹⁴, while delayed developmental milestones are associated with the disorder and predict psychotic symptoms in childhood and adulthood ¹⁵⁻¹⁷. Also, highly heritable neuroanatomical measures, such as cortical thickness (CT) and cortical surface area (SA), show differences between patients with SZ and healthy controls, even prior to the onset of the psychotic symptoms ^{18,19}. All this evidence sustains the prevailing hypothesis that SZ results from environmental and genetic interactions modulating and deviating neurodevelopmental trajectories during the intrauterine and perinatal periods as well as during childhood and early adolescence ^{20,21} that disrupt the ontogenetic plan guiding brain architecture, brain configuration, and brain functioning.”

INTRODUCTION

1) Paragraph 1, Line 32 – does social cognition distinguish humans from other species, or a particular aspect/approach to social cognition?

Social cognition refers to the ability to perceive, understand and interact with others in a social context. It is an important aspect that distinguishes humans from other species in terms of complexity and sophistication, although social cognition can also be observed in other species that display social behaviours.

One of the distinct features of human social cognition is the ability to attribute complex mental states, referred to as Theory of Mind. This involves recognising that others have thoughts, beliefs, desires, and intentions that may not be shared with our own (reviewed in Weng et al., *Neurosci Biobehav Rev* 2022). This allows humans to anticipate and interpret others' behaviours based on their mental states, promoting social interaction, empathy, and cooperation. While some animal species exhibit certain aspects of social cognition, the combination of advanced Theory of Mind, language, and cultural dimensions distinguishes human social cognition as a particularly complex and unique trait (Dumbar, *Annals of Human Biology* 2009, Frith and Frith, *Annual Review of Psychology* 2012).

We have highlighted the human distinctive aspects of social cognition in the introduction section.

Introduction, page 3, lines 30-32: “These symptoms intimately relate the disorder with traits distinguishing humans as a species: abstraction, language, thinking and some aspects of social cognition such as Theory of Mind ¹⁻³.”

2) Paragraph 2, Line 51 – “latter” should be later? This could also be removed altogether.

As we have simplified this section of the introduction, the sentence has changed.

Introduction, page 3, lines 45-47: “Also, highly heritable neuroanatomical measures, such as cortical thickness (CT) and cortical surface area (SA), show differences between patients with SZ and healthy controls, even prior to the onset of the psychotic symptoms ^{18,19}.”

3) Paragraph 5, Line 81 – default mode networks is written out without an acronym in parenthetical notation. The subsequent sentence uses DMN and then authors returns to the full term. Given that “default mode network” seems to only be used a total of three times, I recommend removing the acronym altogether.

We have avoided the acronym use for default mode network and used instead the full spelled term.

Introduction, page 4, lines 73-75: “The same study reveals that the genetic variability in HARs-genes expressed in the brain (HARs-brain genes) is associated with the default mode network functional variation in healthy subjects ⁴⁸.”

4) Line 98 – PRS is mentioned for the first time here; more references to the extant literature should be used in this section.

According to the reviewer’s comment, we have include additional information on polygenic risk score (PRS) and incorporated the most relevant references.

Introduction, page 5, lines 88-96: “Considering the polygenic nature of both cortical structural configuration and SZ’s susceptibility ^{53,54}, studies using measures summarising this complex genetic architecture, such as Polygenic Risk Scores (PRSs), would be helpful to disentangle the genetic roots not only of the disorder but of complex brain traits. The PRS is a quantitative measure of the individual genetic burden for a trait based on Genome Wide Association Studies (GWAS) data. While in psychiatric disorders PRS methods do not yet provide clinically feasible information ⁵⁵, schizophrenia PRSs are highly informative for assessing the individual risk for the disorder at the research level and have been shown to be highly consistent across studies and samples ⁵⁶. The individual PRSs can be calculated at a whole-genome level, but also within subsets of SNPs defined based on their involvement in particular biological pathways of interest ⁵⁷.”

METHODS

Sample

5) Paragraph 1, Line 114 - it is stated that “healthy controls were recruited from the same area.” This refers to Barcelona, Spain? How were HCs recruited and were they compensated?

Yes, the controls were recruited from the same area in Barcelona province from online ads. We have included all this information in the manuscript for clarification.

Methods, Sample section, page 5-6, lines 107-110: “Patients were recruited from inpatient and outpatient units at various centres from the Germanes Hospitalàries located in the Barcelona province. Healthy controls consisted of individuals from the same area, including non-medical staff employed at healthcare facilities, their relatives, and acquaintances, as well as members of the community recruited through online advertisements and independent sources.”

6) Paragraph 1, Line 115 – “patients’ diagnosis” should be diagnoses.

We have corrected the term.

Methods, Sample section, page 6, lines 110-111: “The patients’ diagnoses were confirmed according to DSM-IV-TR based on an interview with two psychiatrists.”

7) Paragraph 1, Line 116 - were all individuals right-handed without reported ambidexterity or nuances? If not, how was laterality determined?

All individuals were unequivocally right-handed based on self-report during the MRI interviews. We have clarified this information in the sample description section.

Methods, Sample section, page 5-6, lines 112-115: “All participants were of European ancestry, between 18 and 65 years old, right-handed (based on self-report) and had an estimated intelligence quotient (IQ) (premorbid IQ in patients), higher than 70, as assessed using the Word Accentuation Test ⁶⁰.”

8) Paragraph 1, Line 119 – did this study take place during the COVID-19 pandemic and, if so, was previous infection considered an exclusionary condition “affecting cognitive or brain function”?

The entire sample was collected before COVID-19 outbreak and thus, all COVID-19 related variables were not assessed.

9) Table 1 – would median be a more appropriate value for illness duration?

We agree with the reviewer that the median would be a more appropriate values for illness duration if the variable would not follow a normal distribution. Nonetheless, after conducting a Kolmogorov-Smirnov test we verified that the variable followed a normal distribution ($D=0.077$, $p=0.074$).

Considering that both the mean and median are nearly the same (illness duration mean=17.30, median=17.19), we have decided to homogeneously define our sample in relation to the demographic and clinical variables, and thus we have left the illness duration description with the mean and standard deviation.

Molecular Analyses

10) Paragraph 2, Line 140 – this is the first mention of SNPs (no literature mentioned in Introduction); though generally understood, this should also be spelled out with acronym separate at first mention.

We have spelled the SNPs acronym at the first mention in the introduction section.

Introduction, page 4, lines 84-85: “In line, subsequent findings also described that Single Nucleotide Polymorphisms (SNPs) in HARs or in linkage disequilibrium with them were more likely associated with the disorder ⁵².”

11) Paragraph 2, Line 144 – it is unnecessary to provide an acronym (e.g., PCA) if the term is used ≤2 times.

We have corrected and omitted the acronym.

Methods, Molecular analyses section, page 7, lines 143-144: “In addition, through a principal component analysis, those individuals found to be related or not of European ancestry were also excluded.”

12) Paragraph 2, Line 144 – how many individuals were excluded due to not being of European ancestry and why was this decision made? This will serve as a limitation.

Out of the initial group of 378 individuals, during the genotyping quality controls it was found that 15 were not of European origin based on the 1000 Genomes data. Considering that the relative allele frequencies in a population and the genetic architecture of an individual greatly depends on the ancestry, population-based genetic studies must be homogenous regarding this variable.

We have discussed the transferability of our findings to other populations of different ethnic origin.

Discussion, page 15, lines 387-391: “Speaking of PRS estimations, our results are based on a sample of European ancestry and SZ GWAS statistics were derived from the European cohort. Then, although GWAS studies performed in non-European samples converge in the same SZ’s genes and pathways ⁹⁸, the extrapolation of our findings to other ethnic groups should not be done straightforwardly and research based on populations of different ethnic origin should be encouraged.”

Surface-Based Morphometry

13) Paragraph 1, Line 195 – in the context of FreeSurfer, how did the authors control for the differences in scanners among individuals (e.g., 1.5T vs. 3T; GE vs. Phillips)? This is noted as a limitation but should be noted in the relevant section that FreeSurfer does not handle this.

We have controlled the scanners among individuals using scanner site as a covariate in all the analyses. Additionally, as suggested by the reviewer we have highlighted in the methods section that structural processing with FreeSurfar was conducted independently for the two scanner sites.

Methods, Surface-based morphometry section, page 8-9, lines 197-200: “The images obtained from the two different scanners were independently pre-processed. This process included the removal of non-brain tissue, an automated Talairach transformation, tessellation of the grey and white matter boundaries and surface deformation ⁶⁸, after which individual images were normalised to a common stereotaxic space.”

Additionally, we highlighted that in order to control for scanner site differences, all the analyses were conducted controlling for this heterogeneity.

Methods, Statistical analyses section, page 10, lines 228-231: “In order to control for the differences in scanners among individuals plus the potential effects of sex, age, premorbid IQ and intracranial volume, all these variables were included as covariates in the analyses.”

RESULTS

PRS Associations with Morphometric Measures

14) Paragraph 3, Line 266 – see Item 11 regarding “Gene Ontology (GO)”.

To clarify the FUMA findings on GENE2FUNC analyses, we have included the figures that were at Supplementary Table 3 in the manuscript main body as Figure 2.

Results, PRS associations with morphometric measures section, page 12: “**Figure 2. Gene set enrichment results.** Functional annotation results in Gene Ontology (GO) categories for biological processes (upper panel) and cellular components (lower panel). In each panel, the GO categories can be seen on the left, followed by the proportion of the overlapping genes and the associated p-value (FDR corrected), and the overlapping genes on the right.”

DISCUSSION

15) Paragraph 1, Line 275 – why do the authors believe that the effect of HARS genetic variability on cortical measures in SZ vs. HC has not been studied previously?

We believe that the scarce research on HARS genetic variability on brain structural phenotypes in schizophrenia is mainly due to the recent emergence of the evolutionary genetics field applied to psychiatric disorders. Following the evolutionary traces of schizophrenia, or other psychiatric disorders, has been challenging, and it has been thanks to the application of comparative genomics tools when advances on the evolutionary forces behind schizophrenia have started to be deeply investigated. Among the relevant evolutionary regions for human-specific evolution, HARS are the ones that have captured much of the attention, especially relative to neurodevelopmental psychiatric disorders and brain phenotypes. There have been studies inspecting HARS variability (relative to sequence and expression variability) in relation to neuroimaging phenotypes, as highlighted in a recent review published by our group (Guardiola-Ripoll and Fatjó-Vilas, International Journal of Molecular Sciences 2023). Such analyses have been focused on the variability in functional networks in healthy controls, or in structural changes shared across different psychiatric disorders (Wei et al., 2019). But we believe that this is soon to be changed.

A proof of interest in this matter, is the recent Science issue dedicated to spread the results of the Zoonomia Project, which highlights the international efforts to unravel the evolutionary forces underlying mammalian evolution. Zoonomia Project features international first-level researchers in comparative genomics and psychiatric genetics, whose joint forces have described the evolutionary pressures contributing to the accelerated evolution of HARS (Keough et al., Science 2023) and highlights the potential of leveraging measures of evolutionary constraint to discover functional variants behind complex human diseases including schizophrenia (Sullivan et al., Science 2023). In line, there is an ENIGMA Evolution working group that seeks to integrate evolutionary genomics findings to understand structural and functional brain variability.

We have modified the sentence in the discussion in order to include and recognise the efforts that have been made in the field of evolutionary genetics applied to psychiatric disorders and encourage future research.

Discussion, page 13, lines 290-293: “Although studies on HARs are an emerging field, as a recent review by our group highlights ⁷⁶, and HARs expression changes have been studied in relation to brain structural changes across different psychiatric disorders ⁴⁸, our study is the first to specifically assess the effect of HARs genetic variability on brain cortical measures in patients with SZ and healthy controls.”

16) Paragraph 4, Line 309 – remain consistent with use of SZ.

We have corrected schizophrenia for SZ term in order to remain consistent with the use of acronyms.

Discussion, page 14, lines 327-329: “First, these findings converge with data highlighting the importance of the foetal period (and the gene expression circumscribed to it) to understand the neurodevelopmental trajectories linked to SZ ¹⁴.”

17) Paragraph 4, Line 313 – use superscript citation for Wei et al. (2019).

We have corrected the citation.

Discussion, page 14, line 341-344: “The authors described those areas of the orbital frontal gyrus and the temporal lobe experienced an x4 and x3 expansion, respectively, and evidenced that the transcription profile of 1711 HARs-genes positively correlated to the pattern of human cortical expansion, meaning that the highest HARs-gene expression occurs in highly expanded areas of the human cortex ⁴⁸.”

18) I recommend discussing future directions stemming from this research.

Following reviewer’s recommendation, we have included an additional paragraph on our perspectives on future research.

Discussion, page 16, lines 397-408: “Our findings, together with the increasing knowledge of the functions of HARs and the biological mechanisms in which they are involved, open new investigation venues. Based on the role of HARs as paramount actors in neurodevelopmental transcriptional regulation and their involvement in the genetic burden of neurodevelopmental psychiatric disorders like SZ and autism ^{42,43,47,82,105,106}, our study and most of the previous research has been focused on HARs common variability. Still, to gain insights into the pathogenic role of these regions, future studies should explore the functional effects of HARs’ rare variability. Also, the prioritization and interpretation strategies in whole-genome sequencing approaches should consider not only exonic or promoter variants but regulatory regions such as HARs. Furthermore, psychiatric disorders with a strong neurodevelopmental component are intimately related to the emergence of the human condition, sustained by the evolution of human-specific ontogenic mechanisms. Thus, studies on the genetic basis of this disorder should not be separated from this evolutionary component. Future studies would greatly benefit from analysing the contribution of evolutionary relevant regions towards the disorders’ genetic background ¹⁰⁷⁻¹⁰⁹.”

REVIEWERS' COMMENTS:

Reviewer #1 (Remarks to the Author):

The authors correctly responded to the reviewer's comments and revised manuscript appropriately. I can agree the publication of revised version of manuscript. I have no further comment.

Reviewer #2 (Remarks to the Author):

The authors have addressed the issues that were raised. The revision has significantly improved the manuscript.

Reviewer #3 (Remarks to the Author):

Revision has fully addressed my concerns.